# Soft Superpixel Neighborhood Attention

**Kent Gauen**
Purdue University
gauenk@purdue.edu

**Stanley Chan**
Purdue University
stanchan@purdue.edu

## Abstract

Images contain objects with deformable boundaries, such as the contours of a human face, yet attention operators act on square windows. This mixes features from perceptually unrelated regions, which can degrade the quality of a denoiser. One can exclude pixels using an estimate of perceptual groupings, such as superpixels, but the naive use of superpixels can be theoretically and empirically worse than standard attention. Using superpixel probabilities rather than superpixel assignments, this paper proposes soft superpixel neighborhood attention (SNA) which interpolates between the existing neighborhood attention and the naive superpixel neighborhood attention. This paper presents theoretical results showing SNA is the optimal denoiser under a latent superpixel model. SNA outperforms alternative local attention modules on image denoising, and we compare the superpixels learned from denoising with those learned with superpixel supervision.[1]

## 1 Introduction

The attention mechanism is attributed to meaningful benchmark improvements in deep neural networks [1]. While the initial attention mechanism acted globally, recent methods operate on local neighborhoods, which requires less computation and reduces the chance of learning spurious correlations [2]. Neighborhood attention (NA) transforms each pixel using its surrounding square neighborhood of pixels. In reality, this square neighborhood is merely an implementation convenience. Natural images contain objects with non-rigid boundaries, such as the contours of a human face or the outlines of text. To account for these deformations, this paper proposes a new local attention module that re-weights the attention map according to low-level perceptual groups, or "superpixels" [3], named soft superpixel neighborhood attention (SNA).

Theory from vision science posits that humans view scenes according to perceptual groups (e.g. Gestalt principles) [4]. Superpixels are one such formulation of perceptual groupings, and they create a deformable, boundary-preserving segmentation of an image [3]. Since deep learning models can learn spurious correlations, operations that utilize perceptual groups should principally help a network learn desirable correlations. Despite the deformable neighborhood shapes, applying attention with a superpixel estimate is straightforward. However, this naive approach is theoretically worse than NA for some superpixel shapes and sensitive to errors in the superpixel estimate. To address these issues, we propose thinking beyond using a single superpixel estimate.

While recent works use the maximum likelihood (ML) superpixel estimate [6, 7], many superpixel formations are equally likely. Figure 1 depicts three samples from a (slightly modified) recent BASS superpixel model [5]. Many of the boundaries seem arbitrary and change among the samples, while some remain fixed. No single superpixel segmentation is the absolute "best". This is not a deficiency of the particular method to estimate superpixels, but rather a drawback of using a point estimate.

---

[1] Code for this project is available at https://github.com/gauenk/spix_paper
[2] For educational value, hyperparameters are modified to create more boundaries.

38th Conference on Neural Information Processing Systems (NeurIPS 2024).

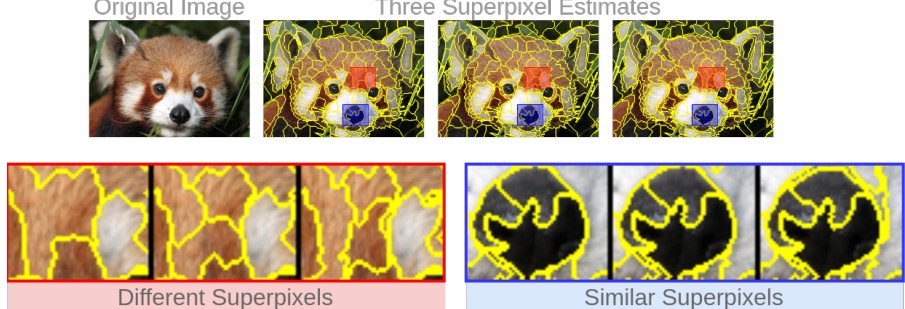

Figure 1: **The Ambiguity of Superpixels.** This figure compares three superpixel estimates from a recent method named BASS. [5][2]While all three samples achieve a similar segmentation quality, some regions are different, and some are almost identical. Since no single segmentation is the "best", this suggests that superpixel assignments are not as important as superpixel probabilities.

This paper presents a theoretically grounded approach to account for the ambiguity of superpixel assignments within an attention module. We consider this ambiguity in our derivation by re-weighting the empirical data distribution with superpixel probabilities. While standard superpixel models assume a fixed prior probability, this paper supposes each pixel has a separate prior superpixel probability. Under this new model, we find the optimal denoiser is the proposed SNA module.

**Contributions.** In summary, this paper presents a rigorous way to incorporate superpixels into an attention module. By modeling image pixels with distinct superpixel probabilities, we find the soft superpixel neighborhood attention (SNA) module is the optimal denoiser. We empirically show the SNA module outperforms NA within a single-layer denoising network by $1 - 2$ dB PSNR. We compare the superpixels learned for denoising with superpixels learned using supervised training [8].

## 2 Related Works

**Non-Local Methods.** Classical image restoration methods transform a pixel using its neighborhood based on pairwise similarities [9–11]. Some early deep learning models incorporate non-local modules with deep learning models for image restoration [12–17]. Once attention was introduced to the computer vision community, later methods drew analogies to between attention and non-local methods [18–20].

**Attention.** While attention modules followed the same form as non-local methods, they were originally presented as a disconnected idea [1, 21]. Attention's primary issue was originally computation, and subsequent research efforts proposed efficient alternatives [22–25, 2]. These alternatives often compute attention across a smaller region, and use a deep network's depth to disperse information globally. Neighborhood attention (NA) is implemented with an efficient, custom CUDA kernel to search a square window and is among the fastest methods [2]. The square window is theoretically more desirable than asymmetrical alternatives, such as SWIN [22].

**Superpixels.** Superpixels embody the concept of perceptual groupings, which suggests that humans first group smaller objects together and then aggregate this information to understand the whole image [3]. There are many methods to model these superpixels [26, 27]. SLIC is a state-of-the-art, classical superpixel method and is explained in Section 3.1 [26]. Superpixels have been applied extensively outside of deep learning literature [28–32]. Superpixels have not been employed as widely since the advent of deep learning, but recent methods have started incorporating them [6, 7]. A recent method, referred to as the superpixel sampling network (SSN), presents a supervised learning method to sample using segmentation labels [8]. The loss function of SSN is related to the idea of superpixel pooling, which we use to evaluate superpixel probabilities in Section 5.3 [33, 34].

**Non-Parametric Density Estimation.** The derivation of the optimal denoiser using latent superpixels closely resembles the problem of mixture-based non-parametric density estimation [35]. This paper does not address important statistical issues, such as identifiability.

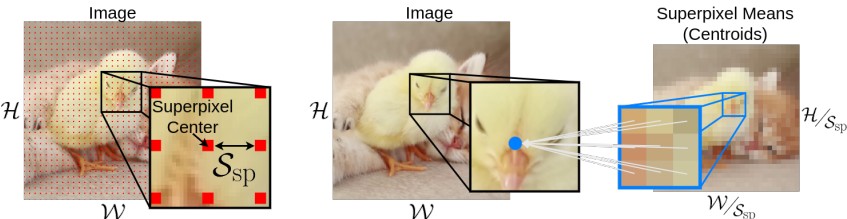

Figure 2: **Each Pixel is Connected to Nine Superpixels.** This figure illustrates the anatomy of the SLIC superpixels. The left-most figure illustrates how superpixels are conceptually distributed across a grid on the input image with stride $\mathcal{S}_{sp}$. The right figure illustrates a single pixel is connected to (at most) nine centroids.

## 3 Preliminaries

### 3.1 SLIC Superpixels

In this paper, an image ($\boldsymbol{x}$) has height ($\mathcal{H}$), width ($\mathcal{W}$), and features ($F$) with shape $\mathcal{H}\mathcal{W} \times F$. Conceptually, superpixels are evenly spaced across the image according to the stride denoted $\mathcal{S}_{sp}$. The superpixel stride ($\mathcal{S}_{sp}$) determines the number of superpixels, $S = \mathcal{H}\mathcal{W}/\mathcal{S}_{sp}^2$ (see Fig 2). SLIC superpixels consist of superpixel means ($\hat{\boldsymbol{\mu}}$ with shape $S \times F$), intra-superpixel precision ($\hat{\boldsymbol{\eta}} = [\hat{\boldsymbol{\eta}}_{app} \ \hat{\boldsymbol{\eta}}_{shape}]$ with shape $S \times 2$), the probability each pixel belongs to each superpixel ($\hat{\boldsymbol{\pi}}$ with shape $\mathcal{H}\mathcal{W} \times S$), and the superpixel assigned to each pixel ($\hat{s}$ with shape $\mathcal{H}\mathcal{W}$). SLIC estimates these quantities using Lloyd's algorithm (similar to k-means) [26]. Following recent works, each pixel is connected to at most nine superpixels [8].

Presently, we describe how SLIC superpixel probabilities are computed. Let the distances between pixels and superpixel means be written as a matrix $\boldsymbol{D}$ of shape $\mathcal{H}\mathcal{W} \times S$ and let $\boldsymbol{D}^{(i,s)} = \|\boldsymbol{x}_i - \boldsymbol{\mu}_s\|_2^2$ if the two are connected and infinity otherwise. A difference between the indices is written $\tilde{\boldsymbol{D}}^{(i,s)} = \|[i_x \ i_y] - [s_x \ s_y]\|_2^2$. Superpixel probabilities are computed as,

$$\hat{\boldsymbol{\pi}}^{(i)} = \sigma(-\hat{\eta}_{i,app}\boldsymbol{D}^{(i)} - \hat{\eta}_{i,shape}\tilde{\boldsymbol{D}}^{(i)}) \tag{1}$$

where $\hat{\eta}_{i,app}, \hat{\eta}_{i,shape} > 0$ enforce consistency across appearance (*app*) and *shape*. The estimated superpixel probabilities are the conditional probability pixel $i$ is assigned to superpixel $s$, $p(s_i = s|\boldsymbol{x}_i) = \hat{\boldsymbol{\pi}}^{(i,s)}$. A superpixel segmentation ($\hat{s}$) is formed by assigning each pixel's cluster label to their most similar superpixel; $\hat{s}_i = \arg\max_s \hat{\boldsymbol{\pi}}^{(i,s)}$. These quantities are *estimates* (note the $\widehat{hats}$) of latent parameters, which is a modeling assumption discussed in Section 4.2.

### 3.2 Superpixel Pooling

The attention module proposed in this paper will learn superpixel probabilities, but standard superpixel evaluation only considers a point estimate. To augment the standard superpixel benchmarks, we follow recent works and compare superpixel probabilities with *superpixel pooling* [8, 5, 34]. A matrix $\boldsymbol{z}$ has size $\mathcal{H}\mathcal{W} \times C$ for $C$ segmentation classes, and superpixel probabilities ($\hat{\boldsymbol{\pi}}$) have size $\mathcal{H}\mathcal{W} \times S$. Then, the probabilities are re-normalized, $\tilde{\boldsymbol{\pi}}^{(i,s)} = \hat{\boldsymbol{\pi}}^{(i,s)}/\sum_{j=1}^{\mathcal{H}\mathcal{W}} \hat{\boldsymbol{\pi}}^{(j,s)}$. Finally, the superpixel pooling operation is written,

$$\tilde{\boldsymbol{z}}_{sp\text{-}pooled} = \hat{\boldsymbol{\pi}}\tilde{\boldsymbol{\pi}}^{\mathsf{T}}\boldsymbol{z} \tag{2}$$

In this paper, the vector $\boldsymbol{z}$ is either an image ($C = F$ is the number of channels) or a segmentation label. A recent method, named the Superpixel Sampling Network (SSN), proposes learning task-specific superpixels with this superpixel pooling loss [8]. Their loss function trains a network to estimate superpixel probabilities by comparing the original and pooled vectors. In Section 5.3, we compare the quality of superpixels learned within an SNA-denoising network and those learned using the SSN loss on the BSD500 dataset [36].

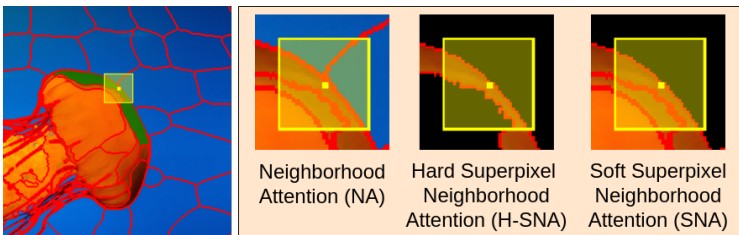

Figure 3: **Superpixel Neighborhood Attention.** The yellow region represents the attention window and the red contours are a superpixel boundary. NA considers all pixels, mixing the dissimilar orange and blue pixels. H-SNA considers only pixels within its own superpixel, which is too few pixels for denoising. SNA excludes the dissimilar blue pixels but retains the similar orange pixels.

### 3.3 Neighborhood Attention

Let a noisy input image be denoted $\boldsymbol{x}$ with shape $\mathcal{HW} \times F$. The queries ($\boldsymbol{q}$), keys ($\boldsymbol{k}$), and values ($\boldsymbol{v}$) project the image $\boldsymbol{x}$ from dimension $F$ to $D$, written as $\boldsymbol{q} = \boldsymbol{x}\boldsymbol{W}_q, \boldsymbol{k} = \boldsymbol{x}\boldsymbol{W}_k, \boldsymbol{v} = \boldsymbol{x}\boldsymbol{W}_v$. Neighborhood attention (NA) computes the output of the attention operator using pixels within a square neighborhood around pixel $i$, denoted $\mathcal{N}(i)$ [2]. A standard neighborhood of size $7 \times 7$ is among the fastest available attention methods. The attention scale controls the shape of the attention weights, $\lambda_{\text{at}} \geq 0$. The output and attention weights are computed below,

$$f_{\text{NA}}^{(i)}(\boldsymbol{x}) = \sum_{j \in \mathcal{N}(i)} w_{i,j} \boldsymbol{v}_j, \qquad w_{i,j} = \frac{\exp\left(\lambda_{\text{at}} d(\boldsymbol{q}_i, \boldsymbol{k}_j)\right)}{\sum_{j' \in \mathcal{N}(i)} \exp\left(\lambda_{\text{at}} d(\boldsymbol{q}_i, \boldsymbol{k}_{j'})\right)} \tag{3}$$

As an additional baseline, this paper augments the standard NA by learning the attention parameter for each pixel. Specifically, each pixel's attention scale is the output of a deep network, $\lambda_{\text{at}}^{(i)} = g_{\phi, \text{Deep}}^{(i)}(\boldsymbol{x})$. Further details are described in Section 4.5.

## 4 Approach

### 4.1 Superpixel Neighborhood Attention

**Hard Superpixel Neighborhood Attention (H-SNA).** Perceptual groupings, such as superpixels, segment pixels into a boundary-preserving partition of the image. These boundaries account for sharp changes in pixel intensity, so using these segmentations as masks can remove perceptually unrelated information. Supplemental Section A.2 formalizes this intuition. H-SNA is a naive implementation of this concept. For a fixed query pixel ($i$), a neighboring pixel ($j$) is masked if its estimated superpixel does not match the query's superpixel, $\hat{s}_i \neq \hat{s}_j$. The function is written as,

$$f_{\text{H-SNA}}^{(i)}(\boldsymbol{x}; \hat{\boldsymbol{s}}) = \sum_{j \in \mathcal{N}(i)} w_{i,j} \boldsymbol{v}_j, \qquad w_{i,j} = \frac{\mathbb{1}[\hat{s}_i = \hat{s}_j] \cdot \exp\left(\lambda_{\text{at}} d(\boldsymbol{q}_i, \boldsymbol{k}_j)\right)}{\sum_{j' \in \mathcal{N}(i)} \mathbb{1}[\hat{s}_i = \hat{s}_{j'}] \cdot \exp\left(\lambda_{\text{at}} d(\boldsymbol{q}_i, \boldsymbol{k}_{j'})\right)} \tag{4}$$

**Soft Superpixel Neighborhood Attention (SNA).** There are two problems with H-SNA. First, H-SNA is unstable when the number of pixels sharing the same superpixel label within the local neighborhood is small. Examples include snake-like tendrils and superpixels that consist of small, disconnected regions (see Section A.2). The second problem with H-SNA is its high sensitivity to improper superpixel estimation. Erroneous ML estimates of superpixels have a dramatic effect on quality. To mitigate both issues, we propose soft superpixel neighborhood attention (SNA), which uses superpixel probabilities instead of superpixel assignments,

$$f_{\text{SNA}}^{(i)}(\boldsymbol{x}; \hat{\boldsymbol{\pi}}) = \sum_{j \in \mathcal{N}(i)} w_{i,j} \boldsymbol{v}_j, \qquad w_{i,j} = \frac{\exp\left(\lambda_{\text{at}} d(\boldsymbol{q}_i, \boldsymbol{k}_j)\right) \sum_{s=1}^{S} \hat{\boldsymbol{\pi}}^{(i,s)} \hat{\boldsymbol{\pi}}^{(j,s)}}{\sum_{j' \in \mathcal{N}(i)} \exp\left(\lambda_{\text{at}} d(\boldsymbol{q}_i, \boldsymbol{k}_{j'})\right) \sum_{s=1}^{S} \hat{\boldsymbol{\pi}}^{(i,s)} \hat{\boldsymbol{\pi}}^{(j',s)}} \tag{5}$$

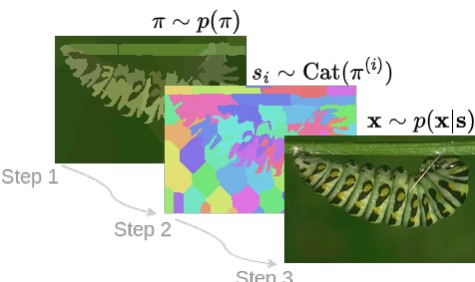

Figure 4: **The Latent Superpixel Model.** The latent superpixel model assumes superpixel probabilities are sampled for each image pixel. This figure illustrates the data-generating process. The leftmost image allows the reader to visually compare the similarities among superpixels by representing each pixel by its most likely superpixel means. Informally, this looks like a "low-resolution image". The superpixels and image pixels are sampled as usual.

Importantly, the sum over all superpixels contains only 9 non-zero terms (see Section 3.1). The shape of the superpixel probabilities allows SNA to interpolate between NA and H-SNA. To be specific, the superpixel probabilities may be flat to match NA, $\hat{\boldsymbol{\pi}}^{(i,s)} \approx \frac{1}{9}$, or the superpixel may be sharp to match H-SNA, $\hat{\boldsymbol{\pi}}^{(i,s)} \approx \mathbb{1}[s = s^*]$. This control over the shape of the superpixel probabilities allows SNA to be better than both H-SNA and NA on average.

## 4.2 The Latent Superpixel Model

Properly incorporating superpixels into attention requires rethinking the relationship between superpixels and images. While superpixels are usually presented with a generative model for images, this model is used only to estimate superpixel assignments. In contrast, this paper uses the data-generating process itself to derive the form of our proposed attention module. Our process is slightly different from standard literature. Ordinarily, all pixels share a single prior superpixel probability. In this paper, the superpixel probabilities are sampled for each image pixel.

For each image pixel, our generative model samples (1) superpixel probabilities $\boldsymbol{\pi} \sim p(\boldsymbol{\pi})$, (2) superpixel assignments $s_i \sim p(s_i|\boldsymbol{\pi}^{(i)})$, and (3) pixel values $\boldsymbol{x}|\boldsymbol{s} \sim p(\boldsymbol{x}|\boldsymbol{s})$. Figure 4 illustrates this proposed latent variable model. Note that SLIC estimated these quantities (recall the hats: $\hat{\boldsymbol{s}}, \hat{\boldsymbol{\pi}}$), but assumes a different data generating model. While the distribution of pixels given the superpixel assignments is unknown, the new model is useful for theoretical analysis. We use it to derive the form of our proposed attention module (Sec 4.3), and to compare the theoretical error between SNA, NA, and H-SNA (Sec A.2).

## 4.3 SNA is the Optimal Denoiser under the Latent Superpixel Model

The optimal denoiser is the function that minimizes the mean-squared error between the denoised pixel and the original pixel value. Following similar derivations [37], minimizing this expectation with respect to an unknown denoiser ($\mathcal{D}$) is possible because of our chosen model for the unknown data density. Say we have a sample, $\boldsymbol{x}, \boldsymbol{s}, \boldsymbol{\pi} \sim p(\boldsymbol{x}, \boldsymbol{s}, \boldsymbol{\pi})$. Then we approximate $p(\boldsymbol{x}_i|\boldsymbol{s}_i)$,

$$\hat{p}(\boldsymbol{x}_i|\boldsymbol{s}_i) = \sum_{m=1}^{M} p(m|\boldsymbol{s}_i)p(\boldsymbol{x}_i|m) = \frac{9}{M} \sum_{m=1}^{M} p(\boldsymbol{s}_i|m)\delta(\boldsymbol{x}_i - \boldsymbol{x}_m) \tag{6}$$

since $p(\boldsymbol{s}_i) = \frac{1}{9}$, and $p(m) = \frac{1}{M}$. By re-weighting with superpixel probabilities, rather than superpixel assignments, our estimate does not depend on ambiguous superpixel labels. The expected loss of a denoiser is computed over the factorized joint density of the Gaussian-corrupted pixels, noise-free pixels, superpixel assignments, and superpixel probabilities: $p(\widetilde{\boldsymbol{x}}_i|\boldsymbol{x}_i)\hat{p}(\boldsymbol{x}_i|\boldsymbol{s}_i)p(\boldsymbol{s}_i|\boldsymbol{\pi}^{(i)})p(\boldsymbol{\pi})$.

---

**Claim 1** *The optimal denoiser to the following optimization problem is soft superpixel neighborhood attention (SNA) when the qkv-transforms are identity, the attention scale is fixed to $\lambda_{at} = \frac{1}{2\sigma^2}$, and the samples used to estimate the data density come from a neighborhood surrounding pixel $i$,*

$$f_{SNA}^{(i)}(\widetilde{\boldsymbol{x}}; \boldsymbol{\pi}) = D^*(\widetilde{\boldsymbol{x}}_i; \boldsymbol{\pi}, \sigma) = \arg\min_{D} \mathbb{E}\left[\|D(\widetilde{\boldsymbol{x}}_i; \boldsymbol{\pi}, \sigma) - \boldsymbol{x}_i\|^2\right] \tag{7}$$

---

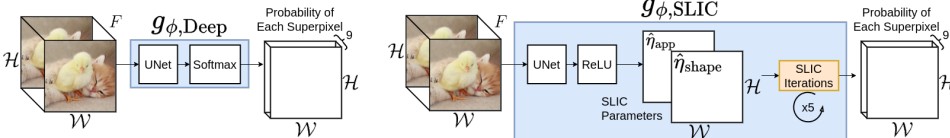

Figure 6: **Estimating Superpixel Probabilities.** The superpixel probabilities are learned during training using one of these two methods. The left method uses a shallow UNet followed by a softmax layer with nine channels ($g_{\phi,\text{Deep}}$). The right method estimates hyperparameters to be used within SLIC iterations ($g_{\phi,\text{SLIC}}$).

See Supplemental Section A.1 for the proof. In practice, the error of a denoiser depends on the sampled data, and in this paper, the data is limited to a neighborhood surrounding the query point. Section A.2 analyzes this limited neighborhood.

## 4.4 Normalization for Restricted Connectivity

SLIC [26] and SNN [8] connect each pixel to (at most) nine superpixels. Therefore, some pixels within a neighborhood may only connect to a subset of superpixels that connect to the query index. This sharp cut-off in connectivity creates artifacts within the attention map since the number of non-zero superpixel probabilities for some neighbors is fewer than nine. These artifacts are due to our choice of superpixel algorithms [26, 8] rather than a limitation of our attention module. To correct these artifacts for the attention map associated with pixel $i$, we re-normalize the adjacent superpixel probabilities to one; $\boldsymbol{\pi}^{(j,s)} \rightarrow \boldsymbol{\pi}^{(j,s)} / \sum_{s' \in \mathcal{C}(i)} \boldsymbol{\pi}^{(j,s')}$, where $\mathcal{C}(i)$ is the set of superpixels connected to pixel index $i$. Figure 5 illustrates the attention maps with and without the normalization step.

$$y_j = \sum_{s=1}^{S} \pi^{(i,s)} \pi^{(j,s)}$$

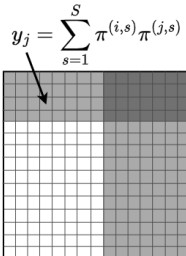

Figure 5: **Artifacts in Attention.** This attention map visualizes the unnormalized superpixel weights for neighbors, $j \in \mathcal{N}(i)$. The restricted connectivity leads to sharp cut-offs in the superpixel weights. To create this figure, $\boldsymbol{\pi}^{(k,s)} = 1/9$ if pixel $k$ is connected to superpixels $s$ and zero otherwise. White pixels indicate the region sums to 1, while grey regions sum to values less than 1.

## 4.5 Estimating Superpixel Probabilities

SNA re-weights the attention map using superpixel probabilities, which are estimated from noisy images, $\boldsymbol{\pi} \approx \hat{\boldsymbol{\pi}} = g_{\cdot}(\boldsymbol{x})$. This paper forgoes considering theoretical concerns (such as identifiability) and uses ad-hoc estimates of these probabilities with two deep networks, depicted in Figure 6.

The first method estimates the marginal probabilities directly from a deep network, $\hat{\boldsymbol{\pi}} = g_{\phi,\text{Deep}}(\boldsymbol{x})$. The second method estimates the superpixel parameters controlling the balance between appearance and shape with a deep network, which is then fed into differentiable SLIC iterations [8] (Section 3.1): $\hat{\boldsymbol{\pi}} = g_{\phi,\text{SLIC}}(\boldsymbol{x}, \hat{\boldsymbol{\eta}})$ where $[\hat{\boldsymbol{\eta}}_{\text{app}} \ \hat{\boldsymbol{\eta}}_{\text{shape}}] = \tilde{g}_{\phi,\text{SLIC}}(\boldsymbol{x})$. This model is abbreviated as $\hat{\boldsymbol{\pi}} = g_{\phi,\text{SLIC}}(\boldsymbol{x})$. The number of SLIC iterations is fixed to 5, following a recent paper [7]. These networks are two-layer UNets [38].

## 4.6 Learnable Attention Scale

Using a network similar to the ones used to estimate superpixel probabilities, all attention modules are augmented to serve as another baseline. The augmentation replaces the fixed attention scale with learnable attention scales for each pixel, $\boldsymbol{\lambda}_{\text{at}} = g_{\phi,\text{Attn}}(\boldsymbol{x})$. The network architecture is identical to $g_{\phi,\text{Deep}}$ except for the final layer, which outputs one channel rather than nine. This baseline empirically demonstrates the theoretical argument from Supplemental Section A.3; even when modulating the attention scale, NA cannot robustly reject dissimilar pixels.

# 5 Experiments

This section demonstrates the impressive benefit of superpixel neighborhood attention compared to standard neighborhood attention. To verify whether the improvement is due to the proposed method, we compare several variations of both attention methods. Section 5.2 compares different attention modules within a simple network architecture on Gaussian denoising, which empirically verifies the theoretical findings in Sections 4.3 and A.2. Section 5.3 compares the superpixel probabilities learned from the denoising loss function with superpixels learned through supervised training. Supplemental Section B.1 includes ablation experiments.

## 5.1 Experimental Setup

Our experiments use a simple denoising network with larger auxiliary networks to highlight the role of the learned superpixel probabilities and attention scale parameters. The denoising network contains only projection layers and a single attention module. To keep the extracted features simple, the only interaction between neighboring pixels is through the attention layer. The project matrices project the input RGB image from three to six dimensions dimension. The full network is written,

$$\hat{\boldsymbol{y}}_{\text{Deno}} = \text{Simple Network}_{\theta,\phi}(\boldsymbol{x}) = f_{\text{Attn}}\left(\boldsymbol{x}\boldsymbol{W}_0, g_\phi(\boldsymbol{x}\boldsymbol{W}_0)\right)\boldsymbol{W}_1 + \boldsymbol{x}\boldsymbol{W}_0\boldsymbol{W}_1 \tag{8}$$

The primary network parameters are denoted $\theta = [\boldsymbol{W}_0, \boldsymbol{W}_1]$, and $g_\phi(\cdot)$ is the auxiliary deep network from Section 4.5. The primary network consists of only 200 parameters ($\theta$), and the auxiliary networks contain about $4.4$k or $8.8$k parameters ($\phi$). We train each network for 800 epochs using a batch size of 2 on the BSD500 dataset [36] using a learning rate of $2 \cdot 10^{-4}$ with a decay factor of $1/2$ at epochs 300 and 600. The network is optimized with Adam [39]. The code is implemented in Python using Pytorch, Numpy, Pandas, and CUDA and run using two NVIDIA Titan RTX GPUs and one RTX 3090 Ti GPU [40–43]. Testing datasets are Set5 [44], BSD100 [36], Urban100 [45], and Manga109 [46].

## 5.2 Gaussian Denoising

This subsection empirically verifies the utility of SNA on Gaussian denoising. The Charbonnier loss is used for denoiser training, $L_{\text{Deno}} = \sqrt{\|\hat{\boldsymbol{y}}_{\text{Deno}} - \boldsymbol{y}\|^2 + \varepsilon^2}$ ($\varepsilon = 10^{-3}$) [47]. By default, the window size is $15 \times 15$, the fixed SLIC precision is ten, and the fixed attention scale is one.

Table 1 presents the quantitative results. The SNA module achieves far better denoising quality than NA, even when NA can learn its attention scale. As an extreme example, the PSNR at $\sigma = 30$ of the left-most SNA column is about **2.8 dB better than standard NA** (2nd NA column). If comparison is restricted to models with the same number of parameters, the PSNR at $\sigma = 30$ of the the fourth SNA column is about **1.8 dB better than the augmented NA** (1st NA column). For a fixed budget of network parameters, SNA yields a higher quality denoiser than NA.

Figure 7 shows SNA is qualitatively superior to NA. The first row shows that SNA produces less grainy images than NA. The bottom two rows show that NA mixes perceptually unrelated information. In the second row, NA outputs a red-orange semi-transparent mist surrounding the orange-red rock. In the third row of the NA results, the white zebra stripes are shaded darker than the clean image when the attention scale is learned, and contain white speckles if fixed. See Section B.2 for more examples.

Learning the attention scale improves standard NA and SNA+$g_{\phi,Deep}$ at higher noise intensities. However, it degrades model quality when superpixels are estimated with SLIC iterations (SNA+$g_{\phi,SLIC}$). Perhaps this quality drop is simply due to network architecture and/or the flow of information. Since the network that learns the attention parameter is only given a noisy image, the simple model struggles to coordinate with the denoising network, which uses multiple SLIC iterations. SLIC iterations explicitly compute pairwise distances, which cannot be learned with the shallow model.

A limitation of the proposed SNA module is the additional computation compared to NA. SNA amounts to re-weighting the attention map produced by a neighborhood search, so SNA is necessarily more compute-intensive than NA. The bottom four rows of Table 1 show SNA is 15 - 22 times slower than NA. While this increase is significant, we believe the numbers reported in this paper are overly pessimistic. Our implementation of SNA has not been optimized, and future development can dramatically reduce the wall-clock time. One way to reduce wall-clock time is to read more efficiently

Table 1: **Image Denoising [PSNR↑/SSIM↑].** The denoising quality of each network is averaged across images from Set5 [44], BSD100 [36], Urban100 [45], and Manga109 [46]. The SNA module is quantitatively superior to NA, even when NA's attention scales are learned with a deep network. However, NA is over 20 times faster than SNA and consumes 13 times less memory. A major contribution of NA is efficiency, while the code for this paper's proposed SNA module has not been optimized. Time and memory usage are reported for a single image of size $128 \times 128$.

| Attn. | SNA | | | | H-SNA | NA [2] | |
|---|---|---|---|---|---|---|---|
| Learn $\lambda_{at}$ | ✓ | ✓ | | | | ✓ | |
| Sp. Model | $g_{\phi,\text{Deep}}$ | $g_{\phi,\text{SLIC}}$ | $g_{\phi,\text{Deep}}$ | $g_{\phi,\text{SLIC}}$ | $g_{\phi,\text{SLIC}}$ | | |
| $\sigma$ | 31.96 | 32.08 | 32.07 | **32.19** | 30.88 | 30.87 | 31.10 |
| 10 | 0.869 | 0.871 | 0.865 | **0.871** | 0.810 | 0.850 | 0.850 |
| 20 | 29.01 | 28.72 | 28.77 | **29.08** | 25.56 | 27.12 | 26.96 |
| | **0.838** | 0.815 | 0.819 | 0.804 | 0.630 | 0.774 | 0.743 |
| 30 | **27.70** | 26.94 | 27.25 | 27.51 | 22.37 | 25.69 | 24.91 |
| | **0.805** | 0.777 | 0.764 | 0.763 | 0.512 | 0.743 | 0.687 |
| Deno Params ($\theta$) | 195 | 195 | 195 | 195 | 195 | 195 | 195 |
| Aux Params ($\phi$) | 8.8k | 8.8k | 4.4k | 4.4k | 0 | 4.4k | 0 |
| Fwd Time (ms) | 30.20 | 45.05 | 27.06 | 40.58 | 28.86 | 4.64 | **2.08** |
| Bwd Time (ms) | 38.72 | 80.93 | 40.00 | 51.35 | 32.54 | 6.06 | **4.67** |
| Fwd Mem (GB) | 1.90 | 2.30 | 1.87 | 2.28 | 1.96 | 0.23 | **0.21** |
| Bwd Mem (GB) | 3.27 | 3.68 | 3.25 | 3.66 | 3.13 | 0.27 | **0.25** |

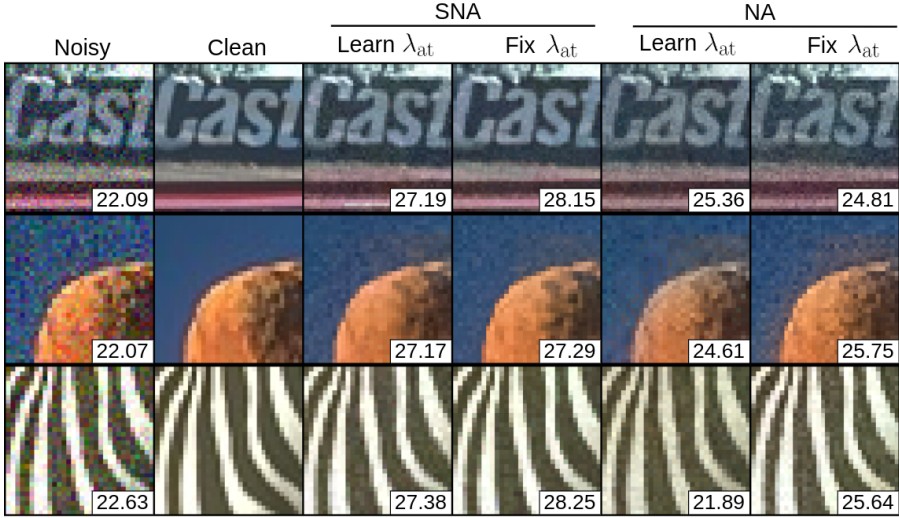

Figure 7: **Denoised Examples [PSNR↑].** This figure compares the quality of denoised images using the Simple Network and noise intensity $\sigma = 20$. The attention scale ($\lambda_{at}$) is either fixed or learned with a deep network. In both cases, the NA module mixes perceptually dissimilar information, while the SNA module excludes dissimilar regions. See Section B.2 for more examples.

from global CUDA memory [43]. Anecdotally, the naive implementations of NA are 500% to 1500% slower than their recently optimized alternatives.

## 5.3 Inspecting the Learned Superpixel Probabilities

This subsection investigates how the superpixels learned from denoising compare with superpixels learned from explicit supervised superpixel training [8]. We observe a collaborative relationship between superpixel probabilities for denoising and superpixel pooling. We observe an adversarial relationship between the superpixels for denoising and boundary adherence. Generally, learning superpixels that improve superpixel benchmarks decreases their utility for denoising.

Table 2: **Supervised Superpixel Training Impacts Denoiser Quality.** This table compares the denoising quality of SNA networks trained with both a denoising loss term and an SSN loss term, $L_{\text{final}} = L_{\text{Deno}} + L_{\text{SSN}}$. The SSN Label "None" indicates only a denoising loss is used. Pixel labels marginally improve the denoising quality, suggesting a cooperative relationship between these optimization problems. Segmentation labels degrade the denoising, suggesting the best superpixels for boundary adherence are not the best superpixels for image denoising. Time and memory usage are reported for a single $128 \times 128$ image.

| SSN Label | PSNR/SSIM | Fwd/Bwd Time (ms) | Fwd/Bwd Mem (GB) |
|---|---|---|---|
| None | 32.77/0.879 | 33 | 0.5 |
| Pix | 32.78/0.889 | 62 | 6.3 |
| Seg | 30.14/0.798 | 86 | 6.3 |

Table 3: **Evaluating Superpixel Quality.** Training an SNA attention module on denoising learns superpixel probabilities with comparable quality to explicitly training superpixels. The ASA and BR metrics evaluate the ML superpixel estimate. The PSNR and SSIM metrics evaluate the quality of the superpixel pooled image.

| SSN Label | | Image Pixels | | | Segmentation | | |
|---|---|---|---|---|---|---|---|
| Loss | $L_{\text{Deno}}$ | $L_{\text{SSN}}$ | $L_{\text{SSN}}$ | $L_{\text{SSN}} + L_{\text{Deno}}$ | $L_{\text{SSN}}$ | $L_{\text{SSN}}$ | $L_{\text{SSN}} + L_{\text{Deno}}$ |
| $\sigma$ | 10 | 0 | 10 | 10 | 0 | 10 | 10 |
| ASA | 0.954 | 0.952 | 0.946 | 0.947 | **0.961** | 0.958 | 0.951 |
| BR | 0.771 | 0.774 | 0.752 | 0.751 | **0.855** | 0.824 | 0.754 |
| PSNR | 27.08 | **31.02** | 29.89 | 29.41 | 21.56 | 22.20 | 25.31 |
| SSIM | 0.811 | **0.934** | 0.886 | 0.883 | 0.553 | 0.574 | 0.737 |

To compare the different training regimes for learning superpixel probabilities, an SNA network is fully trained with a combination of two loss functions. One loss is the denoising loss ($L_{\text{Deno}}$). The second loss is the SSN loss function, which computes the difference between a target vector and its superpixel-pooled transformation (see Section 3.2 for super-pixel pooling) [8]. Specifically, $L_{\text{SSN}} = \mathcal{L}(\hat{z}_{\text{sp-pool}}, z)$ where $\mathcal{L}$ is the cross-entropy loss for a segmentation label and the mean-squared-error for image pixels. The superpixel stride for all methods is fixed to $\mathcal{S}_{\text{sp}} = 14$. Each network estimates superpixels using SLIC iterations, and the attention scale is fixed to one (if applicable). Evaluation is computed on the test set of BSD500 [36].

The Achievable Segmentation Accuracy (ASA) and Boundary Recall (BR) scores evaluate the quality of the maximum likelihood superpixel assignment. The ASA score measures the upper bound on the achievable accuracy of any segmentation method, and the BR measures the boundary alignment between the superpixels and a segmentation. The superpixels are processed with a connected components algorithm before computing ASA and BR scores. To qualitatively compare the learned superpixel probabilities, we follow recent literature and compare the superpixel pooled images [5]. Images are compared against their superpixel pooled images using PSNR and SSIM.

Table 2 evaluates the denoising quality, Table 3 quantitatively evaluates the superpixel quality, and Figure 8 qualitatively evaluates the superpixel pooling quality. We observe a harmony between the denoising loss and pixel pooling loss terms. Combining the two terms yields slightly improved denoising, as reported in the first two rows of Table 2. The first, third, and fourth columns of Table 3 show the ML superpixel quality is similar, but the boundary recall is *slightly worse* when combining the two. The superpixel pooling quality **increases by 2.33 dB PSNR** when combined compared to only denoising. The second and last columns of Figure 8 qualitatively compare the pooled images.

We observe a disharmony between the superpixel probabilities useful for denoising and those trained on segmentation labels. Table 2 reports a **2.6 dB PSNR drop in denoising quality**. The first two rows of Table 3 report training with segmentation labels yields the best ML superpixel estimates, matching related works [8]. However, the superpixel pooling quality is poor. This suggests boundary detection is distinct from denoising and pixel pooling.

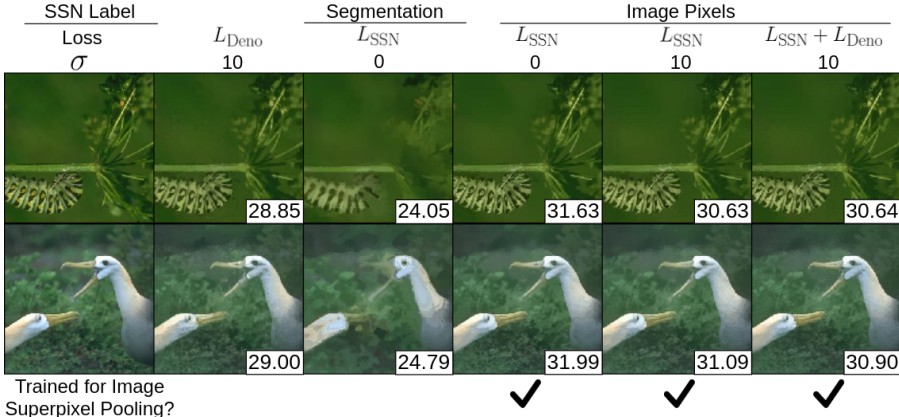

Figure 8: **Comparing Superpixel Probabilites via Superpixel Pooling [PSNR↑].** This figure uses superpixel pooling to qualitatively compare superpixel probabilities learned with different loss functions. Learning superpixel probabilities with only a denoising loss yields better superpixel pooling than supervised learning with segmentation labels. However, jointly training superpixel probabilities for denoising and image superpixel pooling improves denoising and pooling quality, which suggests a useful relationship between the two tasks.

| | SNA | | NA | |
| --- | --- | --- | --- | --- |
| Expression | FLOPs | Memory | FLOPs | Memory |
| $\exp(\lambda_{\text{at}} \boldsymbol{q}_i^\mathsf{T} \boldsymbol{k}_j)$ | $O(F \cdot \mathcal{HW} \cdot K^2)$ | $O(\mathcal{HW} \cdot K^2)$ | $O(F \cdot \mathcal{HW} \cdot K^2)$ | $O(\mathcal{HW} \cdot K^2)$ |
| $\exp(\lambda_{\text{at}} \boldsymbol{q}_i^\mathsf{T} \boldsymbol{k}_j) \sum_{s=1}^S \pi^{(i,s)} \pi^{(j,s)}$ | $O(9 \cdot \mathcal{HW} \cdot K^2)$ | $O(2 \cdot \mathcal{HW} \cdot K^2)$ | | |
| $w_{i,j} = \frac{\exp(\lambda_{\text{at}} \boldsymbol{q}_i^\mathsf{T} \boldsymbol{k}_j) \sum_s^S \pi^{(i,s)} \pi^{(j,s)}}{\sum_{j' \in \mathcal{N}(i)} \exp(\lambda_{\text{at}} \boldsymbol{q}_i^\mathsf{T} \boldsymbol{k}_{j'}) \sum_s^S \pi^{(i,s)} \pi^{(j',s)}}$ | $O(\mathcal{HW} \cdot K^2)$ | $O(\mathcal{HW} \cdot K^2)$ | $O(\mathcal{HW} \cdot K^2)$ | $O(\mathcal{HW} \cdot K^2)$ |
| $\sum_{j \in \mathcal{N}(i)} w_{i,j} \boldsymbol{v}_j$ | $O(F \cdot \mathcal{HW} \cdot K^2)$ | $O(F \cdot \mathcal{HW})$ | $O(F \cdot \mathcal{HW} \cdot K^2)$ | $O(F \cdot \mathcal{HW})$ |
| Total FLOPs & Peak Memory | $O([2F+10] \cdot \mathcal{HW} \cdot K^2)$ | $O(F \cdot \mathcal{HW} + 3 \cdot \mathcal{HW} \cdot K^2)$ | $O([2F+1] \cdot \mathcal{HW} \cdot K^2)$ | $O(F \cdot \mathcal{HW} + 2 \cdot \mathcal{HW} \cdot K^2)$ |

Table 4: **Computational Complexity.** This table reports the FLOPs and memory consumption for Equations 3 and 5. Constant terms are retained to clarify the differences between SNA and NA. The second row re-weights the attention map using superpixel probabilities, and it is the extra step distinguishing SNA from NA. This incurs an extra $O(9 \cdot \mathcal{HW} \cdot K^2)$ FLOPs and requires $O(2 \cdot \mathcal{HW} \cdot K^2)$ memory. However, SNA's additional complexity does not scale with the number of features (F).

### 5.4 Comparing Computational Complexity

Table 4 compares the FLOPs and memory consumption of SNA and NA from Equations 3 and 5. Let the neighborhood window size be written $|\mathcal{N}(i)| = K^2$. In summary, the FLOPs estimate for SNA and NA is $O([2F+10] \cdot \mathcal{HW}K^2)$ and $O([2F+1] \cdot \mathcal{HW}K^2)$, respectively. The peak memory consumption for SNA and NA is $O(\mathcal{HW}F + 2\mathcal{HW}K^2)$ and $O(NF + \mathcal{HW}K^2)$, respectively. Importantly, SNA's additional complexity does not scale with the number of features (F). Since the number of features is a significant factor in the computational cost of an attention operator, it is sensible to believe SNA can be used efficiently within large-scale deep neural networks. To be concrete, if the number of features is 128, then there is less than a 4% percent increase in FLOPs from NA to SNA.

## 6 Discussion

This paper presents the soft superpixel neighborhood attention (SNA) module as an alternative to neighborhood attention (NA). SNA accounts for deformable boundaries of objects. The key modeling assumption is that superpixel probabilities vary per pixel, and we find SNA is the optimal denoiser under this model. For a fixed budget of network parameters, SNA achieves a significantly better denoising quality than NA within a single-layer neural network. While SNA does require more computation than NA, SNA's computation does not scale with features which suggests SNA can be used efficiently within large-scale deep neural networks.

## 7 Acknowledgments

This work is supported, in part, by the National Science Foundation under the awards 2030570, 2134209, and 2133032. The authors thank Drs. Vinayak Rao and David Inouye for their valuable feedback and support.

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

# A  Additional Method Details and Proofs

## A.1  Soft Superpixel Attention is the Optimal Denoiser

The latent superpixel model assumes the following data-generating process for each image pixel,

$$\text{Sample superpixel probabilities} \qquad\qquad\qquad \boldsymbol{\pi} \sim p(\boldsymbol{\pi}) \qquad (9)$$

$$\text{Sample the superpixel assignment given the probabilities} \qquad \boldsymbol{s}_i|\boldsymbol{\pi}^{(i)} \sim p(\boldsymbol{s}_i|\boldsymbol{\pi}^{(i)}) \qquad (10)$$

$$\text{Sample the image pixel given the superpixel assignment} \qquad \boldsymbol{x}|\boldsymbol{s} \sim p(\boldsymbol{x}|\boldsymbol{s}) \qquad (11)$$

We will construct a denoiser that depends on the superpixel probabilities rather than the superpixel assignment. Our derivation is inspired by [37]. Only the form of $p(\boldsymbol{s}_i|\boldsymbol{\pi}^{(i)}) = \text{Categorical}(\boldsymbol{s}_i; \boldsymbol{\pi}^{(i)})$ is known, and $p(\boldsymbol{\pi})$ will not impact our final answer. So we only need to estimate the unknown density, $p(\boldsymbol{x}|\boldsymbol{s})$. To do this, we approximate $p(\boldsymbol{x}|\boldsymbol{s})$ from a sample $(\boldsymbol{x}, \boldsymbol{s}, \boldsymbol{\pi})$,

$$\hat{p}(\boldsymbol{x}_i|\boldsymbol{s}_i) = \sum_{m=1}^{M} p(m|\boldsymbol{s}_i)p(\boldsymbol{x}_i|m) = \frac{9}{M}\sum_{m=1}^{M} p(\boldsymbol{s}_i|m)\delta(\boldsymbol{x}_i - \boldsymbol{x}_m) \qquad (12)$$

where $p(\boldsymbol{s}_i|m) = \boldsymbol{\pi}^{(m,\boldsymbol{s}_i)}$. By re-weighting the empirical density with superpixel probabilities, rather than superpixel assignments, our estimate does not depend on the ambiguous superpixel labels. Notice if the superpixel probabilities are independent of the sample index, then the usual weighting is recovered, $p(m|\boldsymbol{s}_i) = p(m) = \frac{1}{M}$. Next, the pixels are corrupted with Gaussian noise,

$$\hat{p}_\sigma(\tilde{\boldsymbol{x}}_i|\boldsymbol{s}_i) = \left[\hat{p}(\cdot|\boldsymbol{s}_i) * \mathcal{N}(\cdot; 0, \sigma^2\boldsymbol{I})\right](\tilde{\boldsymbol{x}}_i) = \int_{\mathbb{R}^F} \hat{p}(\boldsymbol{x}_0|\boldsymbol{s}_i)\mathcal{N}(\boldsymbol{x}_0 - \tilde{\boldsymbol{x}}_i; 0, \sigma^2\boldsymbol{I})d\boldsymbol{x}_0 \qquad (13)$$

$$= \frac{9}{M}\sum_{m=1}^{M} p(\boldsymbol{s}_i|m)\int_{\mathbb{R}^F}\delta(\boldsymbol{x}_0 - \boldsymbol{x}_m)\mathcal{N}(\boldsymbol{x}_0; \tilde{\boldsymbol{x}}_i, \sigma^2\boldsymbol{I})d\boldsymbol{x}_0 = \frac{9}{M}\sum_{m=1}^{M} p(\boldsymbol{s}_i|m)\mathcal{N}(\boldsymbol{x}_m; \tilde{\boldsymbol{x}}_i, \sigma^2\boldsymbol{I}) \qquad (14)$$

Let the denoiser be a function of the noise intensity and superpixel probabilities, $D(\tilde{\boldsymbol{x}}_i; \sigma, \boldsymbol{\pi})$. The expected error of the denoiser is taken with respect to the density $\hat{p}_\sigma(\tilde{\boldsymbol{x}}_i|\boldsymbol{s}_i)p(\boldsymbol{s}_i|\boldsymbol{\pi}^{(i)})p(\boldsymbol{\pi})$,

$$\mathbb{E}\left[\|D(\tilde{\boldsymbol{x}}_i; \sigma, \boldsymbol{\pi}) - \boldsymbol{x}_i\|^2\right] \qquad (15)$$

$$= \mathbb{E}_{p(\boldsymbol{\pi})}\left[\frac{9}{M}\sum_{s=1}^{S} p(s|\boldsymbol{\pi}^{(i)})\int_{\mathbb{R}^F}\sum_{m=1}^{M} p(s|m)\mathcal{N}(\boldsymbol{x}_m; \tilde{\boldsymbol{x}}_i, \sigma^2\boldsymbol{I})\|D(\tilde{\boldsymbol{x}}_i; \sigma, \boldsymbol{\pi}) - \boldsymbol{x}_m\|^2 d\tilde{\boldsymbol{x}}_i\right] \qquad (16)$$

$$= \mathbb{E}_{p(\boldsymbol{\pi})}\Bigg[\frac{9}{M}\int_{\mathbb{R}^F}\underbrace{\sum_{m=1}^{M}\sum_{s=1}^{S} p(s|\boldsymbol{\pi}^{(i)})p(s|m)\mathcal{N}(\tilde{\boldsymbol{x}}_i; \boldsymbol{x}_m, \sigma^2\boldsymbol{I})\|D(\tilde{\boldsymbol{x}}_i; \sigma, \boldsymbol{\pi}) - \boldsymbol{x}_m\|^2}_{=: \mathcal{L}(D; \tilde{\boldsymbol{x}}_i, \sigma, \boldsymbol{\pi})} d\tilde{\boldsymbol{x}}_i\Bigg] \qquad (17)$$

Note $p(\boldsymbol{s}_i = s|\boldsymbol{\pi}^{(i)}) \to p(s|\boldsymbol{\pi}^{(i)})$ for clarity. The expectation can be minimized by minimizing each integrand for fixed a fixed noisy pixel $(\tilde{\boldsymbol{x}})$ and superpixel probability $(\boldsymbol{\pi})$. Since the integrand is a sum of convex functions, the problem is convex and the solution is the point where the gradient is equal to zero.

$$0 = \nabla_{D(\tilde{\boldsymbol{x}}_i; \sigma, \boldsymbol{\pi})}\mathcal{L}(D; \tilde{\boldsymbol{x}}_i, \sigma, \boldsymbol{\pi}) = 2\sum_{m=1}^{M}\mathcal{N}(\tilde{\boldsymbol{x}}_i; \boldsymbol{x}_m, \sigma^2)\sum_{s=1}^{S} p(s|\boldsymbol{\pi}^{(i)})p(s|m)\left[D^*(\tilde{\boldsymbol{x}}_i; \sigma, \boldsymbol{\pi}) - \boldsymbol{x}_m\right] \qquad (18)$$

In general, the optimal denoiser is written as,

$$D^*(\tilde{\boldsymbol{x}}_i; \sigma, \boldsymbol{\pi}) = \sum_{m=1}^{M} \boldsymbol{x}_m \frac{\mathcal{N}(\tilde{\boldsymbol{x}}_i, \boldsymbol{x}_m, \sigma^2) \sum_{s=1}^{S} p(s|\boldsymbol{\pi}^{(i)})p(s|m)}{\sum_{m'=1}^{M} \mathcal{N}(\tilde{\boldsymbol{x}}_i; \boldsymbol{x}_{m'}, \sigma^2) \sum_{s=1}^{S} p(s|\boldsymbol{\pi}^{(i)})p(s|m')} \qquad (19)$$

By construction, the probability of a particular sample's superpixel assignment is written $p(s|m) = \boldsymbol{\pi}^{(m,s)}$. When the samples are restricted to a neighborhood surrounding pixel $i$, one can write the denoiser as the following expression,

$$D^*(\tilde{\boldsymbol{x}}_i; \sigma, \boldsymbol{\pi}) = \sum_{j \in \mathcal{N}(i)} \boldsymbol{x}_j \frac{\exp\left(-\frac{1}{2\sigma^2}\|\tilde{\boldsymbol{x}}_i - \boldsymbol{x}_j\|\right) \sum_{s=1}^{S} \boldsymbol{\pi}^{(i,s)} \boldsymbol{\pi}^{(j,s)}}{\sum_{j' \in \mathcal{N}(i)} \exp\left(-\frac{1}{2\sigma^2}\|\tilde{\boldsymbol{x}}_i - \boldsymbol{x}_{j'}\|\right) \sum_{s=1}^{S} \boldsymbol{\pi}^{(i,s)} \boldsymbol{\pi}^{(j',s)}} \qquad (20)$$

This is soft superpixel neighborhood attention when the samples are restricted to a neighborhood surrounding pixel $i$, the qkv-transforms are identity, and the attention scale is fixed, $\lambda_{\text{at}} = \frac{1}{2\sigma^2}$.

Presently, we highlight where our per-pixel superpixel probabilities are used in the proof. In Equation 19, the probability $p(s|\boldsymbol{\pi})$ depends on the sampled superpixel probabilities $\boldsymbol{\pi}$. When denoising image pixel index $i$, we have $p(s|\boldsymbol{\pi}) = \boldsymbol{\pi}^{(i,s)}$. However, under the standard model with a shape superpixel probability prior, $p(s|\boldsymbol{\pi})$ does not depend on the particular pixel index. This reduces Equation 20 to a superpixel re-weighting term that does not depend on superpixel information from pixel $i$, which would be strange. We did experimentally try out this alternative re-weighting scheme, and its denoising quality matches NA.

Finally, we discuss a conceptual rationale for sampling superpixel probabilities for each image pixel. The introduction (Sec 1) explains that superpixel assignment is an *ambiguous classification task*. There are many qualitatively "good" superpixel assignments, and often superpixel boundaries can be modified without impacting the overall quality. This suggests superpixel probabilities are more important than superpixel assignments. However, working with $p(\boldsymbol{x}|\boldsymbol{\pi})$ is more challenging than working with $p(\boldsymbol{x}|\boldsymbol{s})$. So in this paper, we still work with $p(\boldsymbol{x}|\boldsymbol{s})$ but design the denoiser to depend on the probabilities $\boldsymbol{\pi}$ to circumvent using the ambiguous superpixel assignments.

## A.2 Expected Denoising Error for a Fixed Neighborhood

Section 4.3 states that SNA is the optimal denoiser, but in practice, samples are limited to a neighborhood surrounding a query pixel. To analyze the impact of this restriction, this subsection computes the expected error over the image pixels and noise for a fixed superpixel sample. For expedient analysis, the number of features is one ($F = 1$, $\boldsymbol{x}_i \to x_i$). The image pixels conditioned on the superpixel information are modeled with a Gaussian distribution, $x_i|s_i, \boldsymbol{\mu}, \boldsymbol{\eta} \sim \mathcal{N}(\mu_s, \eta_s^2)$. We will analyze the attention modules by studying a denoiser of the following form,

$$\mathcal{D}_i(\tilde{\boldsymbol{x}}) = \sum_{j \in \mathcal{N}(i)} w_j \tilde{x}_j \qquad (21)$$

This denoiser corresponds to using a flat attention scale $\lambda_{\text{at}} = 0$ and using only prior superpixel probabilities as attention weights $w_j$. Analysis of attention weights, in general, is difficult, and this approach removes the challenge. The expected error is computed over the Gaussian-corrupted and noise-free data density conditioned by the superpixel information, $p(\tilde{\boldsymbol{x}}|\boldsymbol{x})p(\boldsymbol{x}|\boldsymbol{s}, \mu, \eta)$. The expected error is split into the classic bias-variance decomposition. See Section A.4 for the proof.

**Claim 2** *The expected error of the denoiser $\mathcal{D}$ over the joint density $p(\tilde{\boldsymbol{x}}, \boldsymbol{x}|\boldsymbol{s}, \mu, \eta)$ is written,*

$$\mathbb{E}\left(\mathcal{D}_i(\tilde{\boldsymbol{x}}) - x_i\right)^2 = \sigma^2 \underbrace{\sum_{j \in \mathcal{N}(i)} w_j^2}_{\text{Variance}} + \underbrace{\left[\sum_{j \in \mathcal{N}(i)} w_j(\mu_{s_j} - \mu_{s_i})\right]^2}_{\text{Bias}} \tag{22}$$

The NA module's flat weights achieve the minimum variance of $\frac{1}{W}\sigma^2$ where $W = |\mathcal{N}(\cdot)|$ is the window size. However, NA's bias depends entirely on the superpixel's neighbors and can be large if the neighborhood happens to contain pixels from superpixels with different means. For example, the bias will be large for pixels along the sharp orange edge in Figure 3 since their neighborhoods include the dissimilar blue region. Section A.3 shows that even when the attention scale is non-zero, NA cannot robustly reject dissimilar regions.

H-SNA has the opposite problem of NA. H-SNA's sharp weights achieve the minimum possible bias of zero. However, H-SNA's variance depends entirely on the size of the superpixel. In Figure 3, the snake-like superpixel reduces the number of included pixels, increasing the variance term of the error. This observation that thin and small superpixels are undesirable matches existing assumptions from superpixel literature [26, 5].

SNA's modulation of the superpixel weights allows it to be better than both NA and H-SNA. In Figure 3, SNA excludes the dissimilar blue region (unlike NA) and includes the similar orange regions (unlike H-SNA). Said another way, SNA excludes only dissimilar regions.

### A.3 Exclude Dissimilar Regions Instead of Dissimilar Pixels

Section A.2 showed SNA yields a smaller theoretical error than NA because SNA can exclude dissimilar regions via superpixel probabilities. The previous analysis fixed the attention scale to zero ($\lambda_{\text{at}} = 0$), and one may wonder if the attention scale might be used to exclude dissimilar regions as well. This subsection shows the attention scale excludes dissimilar pixels rather than dissimilar regions, which is less robust under noise. In this analysis, the qk-transforms are fixed to identity, the feature dimension is fixed to 1, and the distances function is the negative squared difference: $d(y_i, y_j) = -(y_i - y_j)^2$.

We first show the attention scale excludes dissimilar pixels rather than regions. Weights of pixels within a similar region should be similar, $w_{i,j} \approx w_{i,i}$, while weights of pixels within dissimilar regions should be near zero, $w_{i,j} \approx 0$. The decay of dissimilar regions will only occur for NA when the scale increases ($\lambda_{\text{at}} > 0$), but this growth also decays the weights associated with similar regions. Noting $e^{\lambda_{\text{at}} d(y_i, y_i)} = 1$, the ratio of two weights for NA is as follows:

$$\log\left(\frac{w_{i,j}}{w_{i,i}}\right) = -\lambda_{\text{at}}(y_i - y_j)^2 = 0 \iff y_i \approx y_j \tag{23}$$

Two noisy pixel values must be approximately equal for their attention weights to be approximately equal. In other words, *excluding dissimilar regions with the attention weights also excludes non-identical noisy pixels.* This strict condition limits NA's ability to exclude dissimilar regions.

Compare this with SNA, which we argue excludes regions more robustly. Say all but a single region should be excluded, so $\lambda_{\text{sp}}$ can be large. The inclusion of the single (useful) region depends on the most likely estimated superpixel matching the latent probabilities, $s_i^* = \hat{s}_i$. Using Section 3.1's formulation, the argmax terms match when the following condition is met,

$$(\hat{\mu}_{s_i^*} - y_i)^2 < (\hat{\mu}_{\hat{s}_i} - y_i)^2, \; s \neq s^* \tag{24}$$

In other words, *SNA properly rejects dissimilar regions when a noisy pixel is more similar to its true mean than another superpixel mean.* Comparing two noisy differences is more robust than comparing two noisy pixels. So we claim SNA rejects dissimilar regions more robustly than NA.

## A.4 The Estimated Expected Denoising Error

This subsection presents a more detailed version of Claim 3 from Section A.2 and its proof.

**Claim 3** *The expected error of the denoiser $\mathcal{D}$ over the joint density $p(\tilde{\boldsymbol{x}}, \boldsymbol{x}|\boldsymbol{s}, \boldsymbol{\mu}, \boldsymbol{\eta})$ is written,*

$$\mathbb{E}\left(\mathcal{D}_i(\tilde{\boldsymbol{x}}) - x_i\right)^2 = \underbrace{\eta_{s_i}^2(1 - 2w_i) + \sum_{j \in \mathcal{N}(i)} w_j^2(\sigma^2 + \eta_{s_j}^2)}_{Variance} + \underbrace{\left[\sum_{j \in \mathcal{N}(i)} w_j(\mu_{s_j} - \mu_{s_i})\right]^2}_{Bias} \quad (25)$$

The following proves the above claim. The first step is to expand the square,

$$\mathbb{E}[(\mathcal{D}_i(\tilde{\boldsymbol{x}}) - x_i)^2] = \mathbb{E}[\mathcal{D}_i^2(\tilde{\boldsymbol{x}})] + \mathbb{E}[x_i^2] - 2\mathbb{E}[\mathcal{D}_i(\tilde{\boldsymbol{x}})x_i] \quad (26)$$

Each expectation is evaluated separately,

$$\mathbb{E}[x_i^2] = \eta_{s_i}^2 + \mu_{s_i}^2 \quad (27)$$

$$\mathbb{E}[\mathcal{D}_i^2(\tilde{\boldsymbol{x}})] = \sum_{j \in \mathcal{N}(i)} w_j^2 \mathbb{E}[\tilde{x}_j^2] + 2 \sum_{k < j, j \in \mathcal{N}(i)} w_j w_k \mathbb{E}[\tilde{x}_j \tilde{x}_k] \quad (28)$$

$$= \sum_{j \in \mathcal{N}(i)} w_j^2 \left(\sigma^2 + \eta_{s_j}^2 + \mu_{s_j}^2\right) + 2 \sum_{k < j, j \in \mathcal{N}(i)} w_j w_k \mu_{s_j} \mu_{s_k} \quad (29)$$

$$= \sum_{j \in \mathcal{N}(i)} w_{i,j}^2(\sigma^2 + \eta_{s_j}^2) + \sum_{j,k \in \mathcal{N}(i)} w_{i,j} w_{i,k} \mu_{s_j} \mu_{s_k} \quad (30)$$

$$\mathbb{E}[\mathcal{D}_i(\tilde{\boldsymbol{x}})x_i] = \sum_{j \in \mathcal{N}(i)} w_j \mathbb{E}[\tilde{x}_j x_i] = w_{i,i} \eta_{s_i}^2 + \sum_{j \in \mathcal{N}(i)} w_{i,j} \mu_{s_i} \mu_{s_j} \quad (31)$$

Then all terms are put together and rearranged,

$$\mathbb{E}[(\mathcal{D}_i(\tilde{\boldsymbol{x}}) - x_i)^2] = \eta_{s_i}^2 - 2w_{i,i} \eta_{s_i}^2 + \sum_{j \in \mathcal{N}(i)} w_{i,j}^2(\sigma^2 + \eta_{s_j}^2)$$
$$+ \mu_{s_i}^2 + \sum_{j,k \in \mathcal{N}(i)} w_j w_k \mu_{s_j} \mu_{s_k} - 2 \sum_{j \in \mathcal{N}(i)} w_{i,j} \mu_{s_i} \mu_{s_j} \quad (32)$$

$$= \eta_{s_i}^2 - 2w_i \eta_{s_i}^2 + \sum_{j \in \mathcal{N}(i)} w_j^2(\sigma^2 + \eta_{s_j}^2)$$
$$+ \sum_{j,k \in \mathcal{N}(i)} w_j w_k \mu_{s_i}^2 + \sum_{j,k \in \mathcal{N}(i)} w_j w_k \mu_{s_j} \mu_{s_k} - 2 \sum_{j,k \in \mathcal{N}(i)} w_j w_k \mu_{s_i} \mu_{s_j} \quad (33)$$

$$= \eta_{s_i}^2(1 - 2w_i) + \sum_{j \in \mathcal{N}(i)} w_j^2(\sigma^2 + \eta_{s_j}^2) + \left[\sum_{j \in \mathcal{N}(i)} w_j(\mu_{s_i} - \mu_{s_j})\right]^2 \quad (34)$$

Using $\mathbb{E}[X^2] = \text{Var}[X] + \mathbb{E}[X]^2$, the bias-variance terms can be identified. Since the $\mathbb{E}[X]^2$ term is simple to compute and exactly matches the right-hand side, we can label those terms as the bias and assign the remaining terms to the variance.

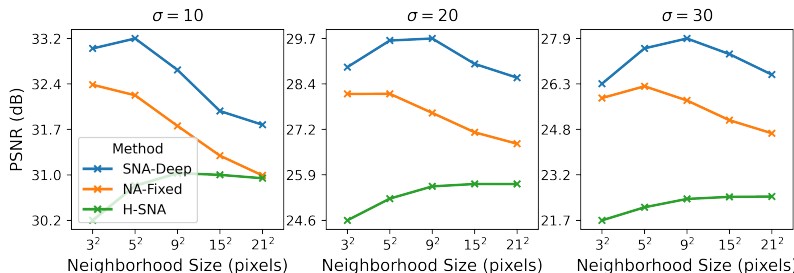

Figure 9: **Neighborhood Window Size.** Increasing the neighborhood window size includes more samples to decrease the variance but also adds bias since the noisy samples can be weighted improperly. This bias-variance trade-off is illustrated by the increasing optimal window size as the noise intensity increases. Since the bias of H-SNA is nearly zero, the increasing neighborhood size only increases the denoising quality within the selected grid.

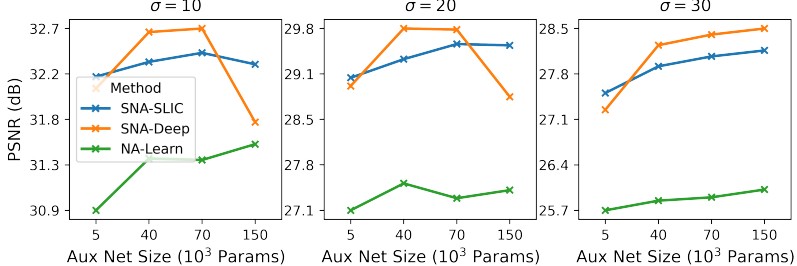

Figure 10: **Number of Auxiliary Parameters.** This ablation experiment expands the size of the auxiliary network by increasing the number of UNet channels. The x-axis plots the number of parameters in the auxiliary network ($|\phi|$). The y-axis plots the PSNR of several denoiser networks. Generally, more parameters improve the denoising quality. The drop in denoising quality for the final network may be due to under-training.

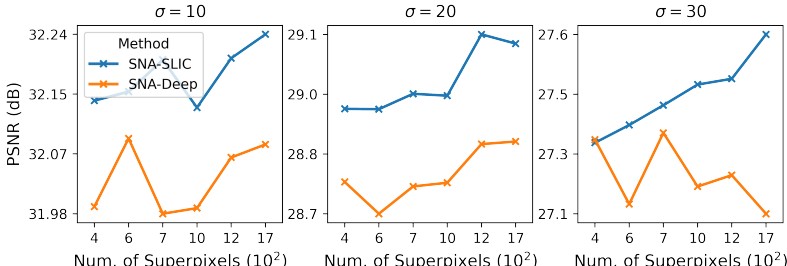

Figure 11: **Number of Superpixels.** The number of superpixels has a limited effect on the denoising quality when compared with other hyperparameters. Generally, increasing the number of superpixels increases the denoising quality. When the noise is low ($\sigma = 10$) the number of superpixels seems irrelevant. As the noise increases, the benefit of more superpixels becomes more apparent. Generally, using explicit SLIC iterations is more effective than predicting superpixel probabilities directly.

# B    Additional Experiments

## B.1    Ablation Experiments

This subsection explores the impact of hyperparameters related to the proposed SNA. This includes the neighborhood window size (Fig 9), the size of the auxiliary network (Fig 10), and the number of superpixels (Fig 11). All networks are trained and tested according to Section 5.2. Both the size of the auxiliary network and the neighborhood window size have a significant impact on the denoiser quality (over 1 dB). The number of superpixels has a smaller overall impact (less than $0.5$ dB).

## B.2 Additional Denoising Results

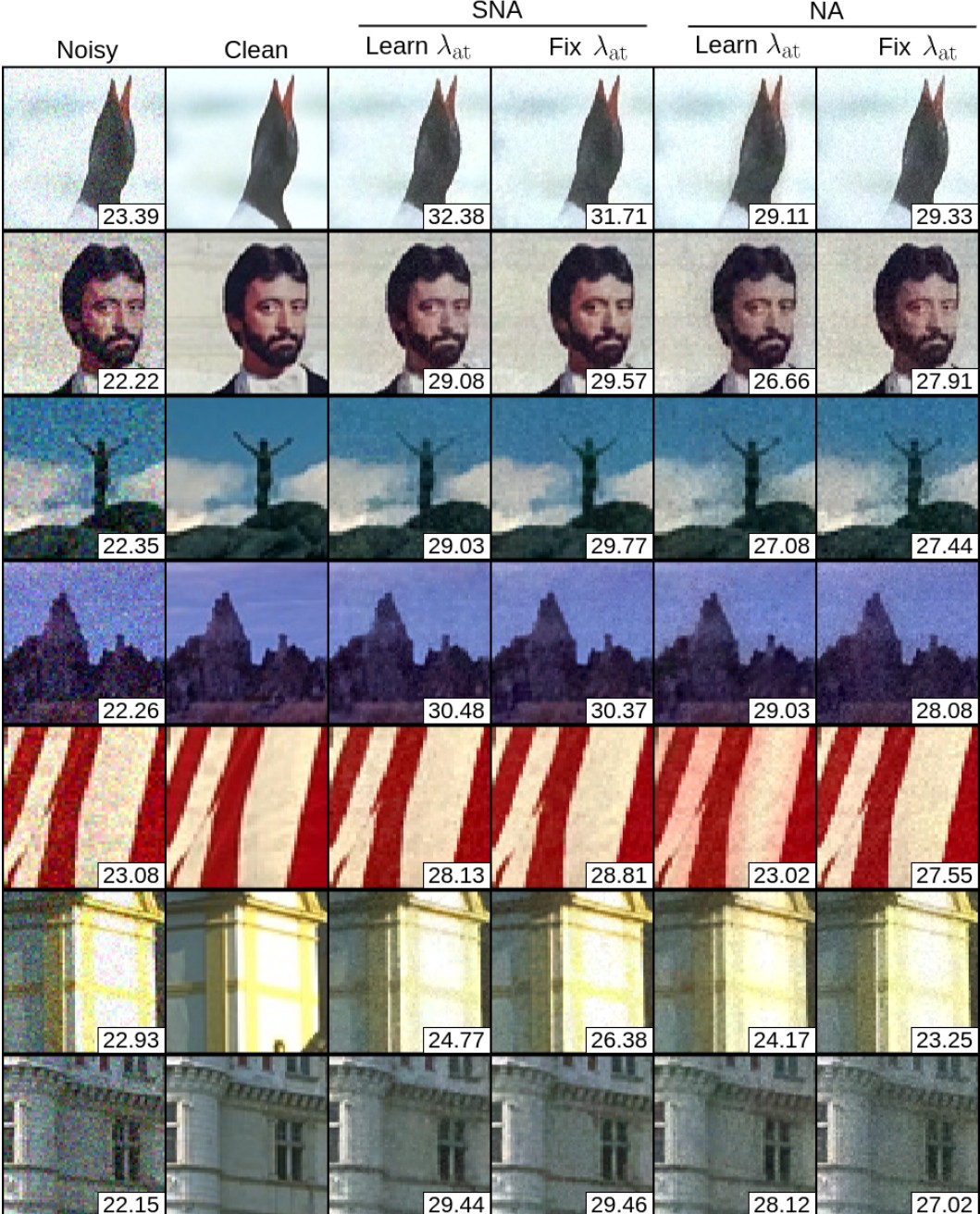

Figure 12: **Additional Denoised Examples [PSNR↑].** This figure compares the quality of denoised images using the Simple Network and noise intensity $\sigma = 20$. The attention scale $(\lambda_{\text{at}})$ is either fixed or learned with a deep network. In both cases, the NA module mixes perceptually dissimilar information, while the SNA module excludes dissimilar regions.

