# OpenReview forum: "Soft Superpixel Neighborhood Attention"
_NeurIPS.cc/2024/Conference — NeurIPS 2024 poster_

### Official Review · Reviewer_dwem · 2024-06-25

**Soundness:** 2
**Presentation:** 2
**Contribution:** 2
**Rating:** 4
**Confidence:** 3

**Summary:**

The paper proposes soft superpixel neighborhood attention, with the motivation that attention is more efficient when it is local, and superpixels are better for local attention as patches, and furthermore, soft superpixels are more robust to errors than hard superpixels.

**Strengths:**

The idea of soft superpixel neighborhood attention makes sense.

Illustrations in the paper are good and appropriate.

**Weaknesses:**

The technical novelty in the paper is not major.

The application (denoising) is not very appealing in the sense that it is not such an important application in the computer vision. It would be best to evaluate the proposed approach on some applications that are more widely used, such as semantic segmentation.

There is a theoretical contribution of optimality of the proposed denoiser is proven, but it assumes the proposed latent superpixel model, and it is not clear how true of the reality this proposed model is.

**Questions:**

see above

**Limitations:**

Limitations are addressed.

---

> ### Author Rebuttal · Authors · 2024-08-06
>
> Thank you for your thoughtful comments. We are encouraged to learn that you find our proposed method makes sense and that our illustrations are good and appropriate.
>
> We are disappointed to learn you find image denoising not a significant enough task for publication.
>
> We are also disappointed to learn you believe “the technical novelty of the paper is not major.” This belief appears to stem from your comment expressing concern about the reality of a latent superpixel model. In general, modeling assumptions do not match reality, so we do not understand why you find non-realistic modeling assumptions troubling. And without a reference, we are unsure why you find this paper lacking novelty.

---

### Official Review · Reviewer_nsbe · 2024-07-08

**Soundness:** 4
**Presentation:** 4
**Contribution:** 4
**Rating:** 7
**Confidence:** 4

**Summary:**

The paper proposes a new attention mechanism based on Soft Superpixel Attention. The main idea is to use superpixel segmentations directly in the attention module. The paper proposes rigorous proof that the proposed mechanism is the optimal denoiser. Results show an improved denoising performance.

**Strengths:**

* The paper presents an elegant and novel idea to use superpixels in the attention module.
* It is well-written with clear explanations. I appreciate that parts that are less important for paper understanding are moved to the supplementary materials (e.g. proofs).
* The experiments are sufficient with an improved performance over the state-of-the-art.

**Weaknesses:**

* The performance improvement is relatively small.
* The paper would be much stronger if it could show how this attention module could be used in other tasks than only on denoising. For instance, deblurring, object detection, tracking, etc.

**Questions:**

How difficult is it to incorporate the proposed attention mechanism in other downstream tasks?

**Limitations:**

Yes, this is discussed.

---

> ### Author Rebuttal · Authors · 2024-08-06
>
> Thank you for your comments. We are grateful for your positive feedback.
>
> **Q1.** In principle, SNA can directly replace (or augment, depending on your perspective) current NA modules for applications beyond image denoising. The practical restriction is our prototype (and slower) implementation of SNA, which makes training/fine-tuning networks needlessly expensive. Please see our “Global Rebuttal” for more details.

---

> > ### Comment · Reviewer_4VnB · 2024-08-12
> >
> > Thanks for your response.
> >
> > I maintain my previous score of "accept".

---

> > ### Comment · Reviewer_nsbe · 2024-08-13
> > **Response**
> >
> > Thank you for the rebuttal. I maintain my accept rating.

---

### Official Review · Reviewer_4VnB · 2024-07-09

**Soundness:** 3
**Presentation:** 3
**Contribution:** 3
**Rating:** 7
**Confidence:** 4

**Summary:**

This paper proposes a soft superpixel neighborhood attention (SNA). It proves that SNA is the optimum denoiser under Gaussian noise. Experiments show that SNA outperforms other local attention modules for the image denoising task.

**Strengths:**

- This is an interesting theoretical study, backed up with experiments.
- It is refreshing to see some theoretical work submitted to Neurips
- Using non-rigid boundaries instead of square neighborhood makes sense.
- Results are promising

**Weaknesses:**

- Experiments are limited. The network that has been tested is written in Eq. (10).
- The current implementation is slow. Some effort is needed to program an efficient version, but this is left for future work.

**Questions:**

I am not sure that I understand the training process. It is claimed that the network is trained on BSD500, but do you train by adding noise to the image, and targeting the network to produce the clean, original image?

**Limitations:**

The hypothesis for the theory are clearly stated, and the limitation are acknowledged.

---

> ### Author Rebuttal · Authors · 2024-08-06
>
> Thank you for your comments. We are encouraged by your positive feedback. We acknowledge our current prototype implementation of SNA is slower than NA. Your comment is similar to reviewer XzEJ’s comment about computational complexity, so we address this weakness in the “Global Rebuttal.”
>
> **Q1.** Your summary is correct. Gaussian noise is added to an image which is fed into the network. The network’s output is a denoised image, which is compared to the clean image using the Charbonnier loss (see the inline equation on line 191).

---

### Official Review · Reviewer_XzEJ · 2024-07-14

**Soundness:** 4
**Presentation:** 3
**Contribution:** 4
**Rating:** 7
**Confidence:** 5

**Summary:**

This paper proposes an attention module in which the dot product weights are modified with
superpixel probabilities, named Superpixel Neighborhood Attention (SNA). By doing so, the
optimization process is arguably made easier by letting attention avoid learning spurious
interactions, which prior work into the attention module has found to be somewhat common.
One key difference to prior works in superpixels is modelling superpixels as probabilities instead
of naive binary interactions, which in theory allows a flexible interpolation between standard
attention and superpixel attention. The proposed module is shown to improve the baseline approach
without superpixels by around 1 to 3 dB PSNR, and qualitative results support this claim as well.

**Strengths:**

1. Unique approach for injecting superpixel information directly into an attention operator. From
my point of view, this is among the most significant contributions of the paper, which would enable
improved achievable performance with attention in the context of larger resolution problems, of
which denoising is one. I think the impact of this approach extends beyond denoising and super
resolution applications.

2. Significant improvement from the baseline approach, neighborhood attention. By allowing
irrelevant local interactions to be avoided, SNA provides considerably better denoising quality than
the standard neighborhood attention.

**Weaknesses:**

1. From my point of view, the most considerable weakness is a lack of discussion on the
computational complexity of the proposed approach. Note that I am not referring to the reported
performance gap, but generally to the fact that it is unclear from reading the paper how complex the
SNA attention algorithm is, and whether or not it can scale beyond the scope of the experiments done
in this paper. While metrics such as time and space complexity, and similar metrics such as FLOPs
don't necessarily translate into actual efficiency, they do provide a rough idea of whether or not
two operations are at least comparable in terms of resource requirement in theory. Given that the
baseline, neighborhood attention, is seemingly so much more efficient in runtime and memory usage,
and the fact that it has been performance optimized to some extent over time, and that these are
both operations in complex deep neural networks implemented primarily by a deep learning framework
with a different standard of implementation, I think the reported numbers are simply not that
informative. The disclosure and clarity is certainly appreciated, but stating that the proposed
approach's performance levels are "likely overly pessimistic" is not informative, and could be
easily reworded by providing some evidence that there isn't such a significant difference in
performance levels in theory.

2. Lack of comparison to other relevant approaches. I think limiting the use of dot product
attention to just one local pattern, namely neighborhood attention, is worth a discussion. Is there
a reason why self attention itself is not considered here, given that the superpixel probabilities
effectively serve as an implicit attention mask. It may also be true that one can map superpixel
probabilities directly to attention weights and define attention differently. I think while the
paper is mostly complete as is, this is something that merits at least a paragraph.

3. The citation for neighborhood attention is incorrect; the reference ("Neighbor2neighbor") does
not mention neighborhood attention. The correct reference is:

>Hassani, A., Walton, S., Li, J., Li, S. and Shi, H., 2023. Neighborhood attention transformer.
>In Proceedings of the IEEE/CVF Conference on Computer Vision and Pattern Recognition (pp. 6185-6194).

**Questions:**

1. Is there any metric through which SNA's cost or resource requirement can be compared against
standard neighborhood attention with more certainty? The seemingly large performance gap in terms of
actual runtime and memory footprint, while necessary, is unsurprising, and only really reveals that
as part of future efforts, SNA could use performance optimization. However, it is completely unclear
how far that optimization could go. My suggestion is at the very least providing some other proxy
for that, be it in the form of time and space complexity, or any other relevant analysis of the
steps required by the algorithm and how easy / difficult they will be to parallelize and performance
optimize.

2. Can you elaborate Eq (1)? Softmax is an operation over a set or vector, and I would assume it
would be over N values per pixel, assuming each pixel has N superpixels.

3. This is just out of curiosity, but as of recently the neighborhood attention package does support
non-square kernels. Does that change in assumptions with regard to neighborhood attention change
anything about the significance of SNA over NA, or is that just a minor detail?

4. How are the efficiency performance metrics (runtime and memory usage) evaluated? Were standard
practices followed? (i.e. locking power and frequency, benchmarking without external factors such as I/O,
communication, framework-level optimizations such as caching allocators and memory planning,
measuring metrics iteratively and reporting an average, and the like)

5. Do either of SNA / H-SNA use the original neighborhood attention implementation, or is the
implementation done in this paper completely specific to superpixels? I'm trying to understand
whether SNA or its future application could also use additional performance optimizations done to
neighborhood attention, or whether it would require its own independent development and in turn face
similar issues in terms of extensibility as any other such approach.

**Limitations:**

While the reported performance gap is a limitation of the proposed approach at the moment, I think
it is very insignificant when taking into account the rest of the contributions of the paper.

However, I cannot be sure of that without evidence that would suggest the performance gap is simply
due to a lack of performance optimization. It could very well be the case that SNA in its current
form cannot scale easily beyond the scope of this paper, which would be in my opinion a far greater
limitation.

This is therefore either a limitation (meaning it is simply not possible or extremely difficult to
analyze SNA's computational cost and compare against standard neighborhood attention) or a weakness
(it was just not included in the paper.)

---

> ### Author Rebuttal · Authors · 2024-08-06
>
> Thank you for your comments. We are encouraged by your positive feedback. Thank you for pointing out the incorrect citation; we will update the reference in our paper. We note that we do not compare with self-attention because the computational complexity of a global search makes self-attention impractical.
>
> **Q1.** Thank you for this detailed comment. We agree a more concrete comparison is needed. Since your comment is similar to reviewer 4VnB’s comment about SNA's implementation, we address this concern in more detail in the “Global Rebuttal”. In summary, we find SNA’s additional complexity does not scale with the number of features (F). This suggests SNA can be used efficiently within deep neural networks, since the number of features is a major factor of a network’s overall computational burden.
>
> **Q2.** Your summary is correct. The softmax expression in Equation 1 is applied over the superpixels connected to pixel i.
>
> **Q3.** I would not expect the non-square kernels of NA to significantly change this paper’s findings. Using the non-square kernels of NA would be similar to Hard-SNA. However, the non-square NA is restricted to rectangular regions rather than Hard-SNA’s deformable regions.
>
> **Q4.**  To evaluate runtime and memory usage, no other processes on the computer are running. The procedure is executed for each operator (SNA or NA), and the reported times are averaged over three runs. No major IO communication is included in the timed section of the program. Our procedure is as follows:
> ```
> a. The CUDA context is initialized with a non-measured CUDA function.
> b. The CUDA memory usage is reset and the CUDA cache is emptied.
> c. The CUDA device is synchronized with the host
> d. A timer is started
> e. The operator of interest (SNA or NA) is executed
> f. The CUDA device is synchronized with the host
> g. The timer is stopped. The time difference and peak memory usage are recorded
> ```
>
> We are open to modifying our benchmark procedure. For example, we will empty the CUDA Caching Allocator in step (b) by executing a `torch.cuda.memory.empty_cache()` command for our Pytorch script.
>
> **Q5.** Our implementation of SNA uses two operators within the NA package. First, the attention map is computed using code from NA. Second, the attention map is then re-weighted using SNA's superpixel weights. Third, the attention map is aggregated using code from NA. The dependency of SNA on NA modules suggests that SNA's performance will improve as NA's performance improves.

---

> > ### Comment · Reviewer_XzEJ · 2024-08-10
> >
> > Thank you; my questions have been resolved and I don't have any more.
> > I'm changing my confidence score and still vote for this paper to be accepted.

---

### Author Rebuttal · Authors · 2024-08-06

**Summary.**

We thank the reviewers for their thoughtful feedback. We are encouraged by the positive comments.

Two reviewers comment favorably on the novelty of the paper. One reviewer states the approach is “unique” [XzEJ] and another reviewer states the idea is “elegant and novel” [nsbe]. Of the remaining two reviewers, one states the paper is an “interesting theoretical study” [4VnB] and both state our proposed method “makes sense” [4VnB,dwem]. One reviewer, dwem, claims “the technical novelty in the paper is not major”. However, dwem provides no direct explanation for their opinion and does not provide an alternative reference.

Three reviewers find the experimental results sufficient for publication. One reviewer notes our method is a “significant improvement from the baseline approach” [XzEJ]. Reviewer 4VnB states the “results are promising”, and reviewer nsbe states the “experiments are sufficient”.

**Compute Complexity.**

There is interest about the compute complexity of SNA. XzEJ states “the most considerable weakness is a lack of discussion on the computational complexity of the proposed approach”. A related comment from 4VnB states the “current implementation is slow”. To address these concerns about computation, we provide an analysis of SNA’s compute complexity in the attached pdf.

Table 1 in the attached pdf breaks down the computational complexity of each step in NA and SNA (equations 3 and 5). In summary, the FLOPs estimate for NA is O($[2F + 1]\cdot NK^2$) and for SNA it is O($[2F + 10]\cdot NK^2$). The peak memory consumption for NA is O($NF + NK^2$) and for SNA it is O($NF + 2\cdot NK^2$). Importantly, SNA’s additional complexity does not scale with the number of features (F). Since the number of features is a significant factor in the computational cost of an attention operator, it is sensible to believe SNA can be used efficiently within large-scale deep neural networks. To be concrete, if the number of features is 128 then there is less than a 4% percent increase in FLOPs from NA to SNA.

We will include Table 1 in the supplemental section of the paper.

**Why is SNA Only Demonstrated on Image Denoising?**

XzEJ notes “the impact of this approach extends beyond denoising and super-resolution applications”. This sentiment is echoed by nsbe and dwen, who express interest in applications of SNA beyond image denoising. They recommend incorporating SNA into deep neural networks for other computer vision tasks, such as “deblurring, object detection, tracking” [nsbe] and “semantic segmentation” [dwem]. Reviewer nsbe asks directly: “How difficult is it to incorporate the proposed attention mechanism in other downstream tasks?”

In principle, SNA can directly replace (or augment, depending on your perspective) current NA modules. The practical restriction is our slower, prototype implementation of SNA. This proof-of-concept implementation makes the already expensive task of training/fine-tuning needlessly more expensive. We believe properly optimizing SNA’s implementation is a worthy, separate research effort, and this optimization should be completed before investing resources into large-scale training. By demonstrating a significant improvement on image denoising, we hope to stimulate the community to work with us and make SNA possible for other tasks.

---

### Comment · Area_Chair_CZ2F · 2024-08-08
**Paper discussion and rating finalization**

Dear Reviewers,
Can you please have a look at the reports of the other reviewers and also the rebuttal from the authors and respond to their questions, if available, then discuss any issues of your concern, finalise and reach a consensus about the rating of the paper before the deadline of the next Tuesday, 13th August?
Thank you very much for your time, effort and contribution to the organization of NeurIPS 20024,
AC

---

### Comment · Area_Chair_CZ2F · 2024-08-11
**Reminder of the deadline**

Dear Reviewers,
While the deadline, Tuesday, 13th August, is approaching, can you please check at your early convenience the rebuttal from the authors, make or require further clarifications, if necessary, and interact with the authors over any further issues/concerns you may still have, and finalise the rating of the paper soon. Even though you have no further issues/concerns, you may want to acknowledge the responses to your questions in the rebuttal from the authors.
Thank you very much for your time, effort and contribution to the organization of NeurIPS 2024,

---

### Decision · Program_Chairs · 2024-09-25

**Decision:**

Accept (poster)

**Comment:**

The paper has received three strong support, but one borderline reject rating that has concerns about the technical novelty, significance and impact of the topic. But the reviewer did not elaborate the concerns and interact with the authors. More discussions, explanations and clarifications have been provided to largely the satisfactions of the relevant reviewers.